# MODEL-HETEROGENEOUS FEDERATED PROMPT LEARNING

## ABSTRACT

Large-scale vision-language models (VLMs) have shown remarkable transferability across tasks, and their synergy with federated learning (FL) frameworks offers promising privacy-preserving learning capabilities. Recent advances in federated prompt learning (FPL) leverage prompt tuning to reduce the computation and communication overhead. However, existing FPL methods assume a homogeneous model setting, where all clients share the same VLMs, which is impractical given the heterogeneous computational capacities of clients in real-world scenarios. To bridge this gap, we propose model-heterogeneous federated prompt learning (MHFPL), a novel setting where clients with diverse VLM backbones collaboratively learn prompts. We further introduce `FedAPPR`, a principled framework for MHFPL built on two key components: (a) server-level adversarial prompt alignment for aligning client semantics via adversarial training, and (b) client-level proximity regularization to further constrain prompt drift between clients. Extensive experiments on six datasets with diverse architectures and data distributions demonstrate the superiority and generalization of `FedAPPR`, confirming it as an effective solution for FL with varying VLMs.

## 1 INTRODUCTION

Recent advances in large-scale vision-language models (VLMs), such as CLIP (Radford et al., 2021), ALIGN (Jia et al., 2021), and Flamingo (Alayrac et al., 2022), have exhibited remarkable generalization abilities, allowing them to perform effectively across a wide range of downstream tasks. Integrating these powerful pre-trained models into the federated learning (FL) paradigm (McMahan et al., 2017; Yang et al., 2019) opens up exciting possibilities for privacy-conscious collaborative learning, where decentralized clients benefit from joint model training without exchanging raw data. Efforts to leverage vision transformers (Dosovitskiy et al., 2021) in FL settings have shown promising results in enhancing robustness under non-i.i.d. data conditions (Qu et al., 2022; Sun et al., 2023). Despite their potential, the deployment of such large-scale architectures in practical FL environments is restricted by substantial computational burdens and high communication costs (Yang et al., 2023), which hinder their scalability and practical applicability.

To mitigate these challenges, prompt learning (PL) (Zhou et al., 2022b; Liu et al., 2023) has emerged as a lightweight alternative for adapting pre-trained models. By freezing the backbone and tuning only a small set of prompt parameters, PL drastically reduces the training footprint. Extending this concept, PromptFL (Guo et al., 2024) introduces the federated prompt learning (FPL) framework that integrates PL with the FedAvg (McMahan et al., 2017) protocol. In this framework, each client locally updates the prompt vectors using their private data, which are then aggregated on the server to form a global prompt representation. Since only prompts are updated and exchanged, PromptFL significantly lowers both computation and communication demands. Building on this foundation, a growing body of work has proposed personalized prompt strategies to address client-specific data disparities (Yang et al., 2023; Guo et al., 2023; Li et al., 2024; Cui et al., 2024; Luo et al., 2025), improving adaptability in heterogeneous federated scenarios.

However, existing FPL methods operate under a homogeneous model assumption, i.e., *all clients use the same VLM architecture* for prompt learning. This assumption is unrealistic in practice. In real-world scenarios, client devices vary widely in computational resources, operating environments, and pre-installed model architectures. In practice, clients may employ different VLMs tailored to their local constraints (e.g., ViT or ResNet),

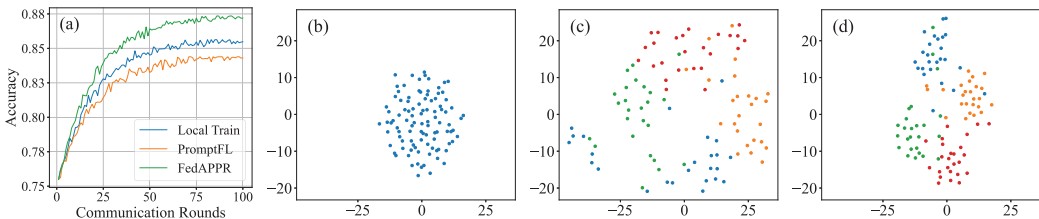

Figure 1: (a) Accuracy of Local Train, PromptFL, and FedAPPR vs. rounds. PromptFL performs worse than Local Train, while FedAPPR achieves better results. (b) Visualization of prompts learned by PromptFL across clients under model-homogeneous settings, where each point represents a client and prompts are well-aligned. (c)/(d) Visualization of prompts learned by PromptFL/FedAPPR under model-heterogeneous scenarios, where colors indicate clients with different model architectures. PromptFL yields scattered and inconsistent prompts across clients, while FedAPPR produces more coherent and structured ones. Implementation details are in the Appendix.

resulting in varying model capacities and prediction accuracies, as illustrated in Figure 2. Consequently, enforcing model uniformity is often infeasible or counterproductive. Furthermore, this model heterogeneity can lead to significant prompt divergence in the federating process, as shown in Figure 1 (b) and (c). This divergence presents significant challenges for prompt aggregation and knowledge sharing across clients, as depicted in Figure 1 (a), where PromptFL performs worse than Local Train.

To bridge this gap, we introduce a novel and practical problem setting: *model-heterogeneous federated prompt learning (MHFPL)*, where each client retains its own VLM and collaboratively learns prompt representations through federated communication. To tackle the MHFPL problem, we propose the Federated Adversarial Prompt Alignment and Proximity Regularization (FedAPPR), a principled framework for MHFPL that introduces two key mechanisms: a) adversarial prompt alignment: An architecture-agnostic alignment strategy that encourages prompt distributions from different clients to share more coherent semantics through adversarial learning; b) proximity regularization: A lightweight regularization term that pulls each local prompt toward a global reference, further reducing inter-client divergence. The whole FedAPPR framework is illustrated in Figure 3, and our key contributions are summarized as follows:

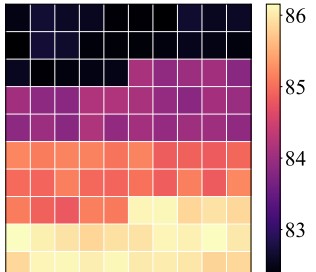

Figure 2: Client-wise accuracy across 100 clients, grouped into four sets of 25 based on different VLM backbones (e.g., first 25 blocks correspond to first backbone). Clear inter-group performance disparities reveal notable model heterogeneity. Implementation details are in Appendix.

* ⋆ We focus on an important and practical FPL problem, where clients deal with varying model architectures. We formalize this problem as a new learning topic called model-heterogeneous federated prompt learning (MHFPL) and introduce the FedAPPR method to approach it.
* ⋆ FedAPPR integrates two complementary strategies: An adversarial alignment mechanism that enforces cross-client semantic consistency by training prompts to fool a server discriminator, making them architecture-invariant; A proximity regularization term that softly encourages each local prompt to stay close to a global reference prompt, further reducing variance.
* ⋆ Experiments on six benchmarks validate the superior performance of FedAPPR over baselines, demonstrating its effectiveness in handling heterogeneous backbones and highlighting its potential for federated learning with diverse VLMs.

## 2 RELATED WORK

### 2.1 MODEL-HETEROGENEOUS FEDERATED LEARNING

Federated learning (FL) (McMahan et al., 2017; Yang et al., 2019) has emerged as a prominent distributed learning paradigm that addresses privacy and security concerns by allowing collaborative model training among decentralized clients. However, a fundamental challenge in FL arises from model heterogeneity across clients, which is often caused by variations in local computational ca-

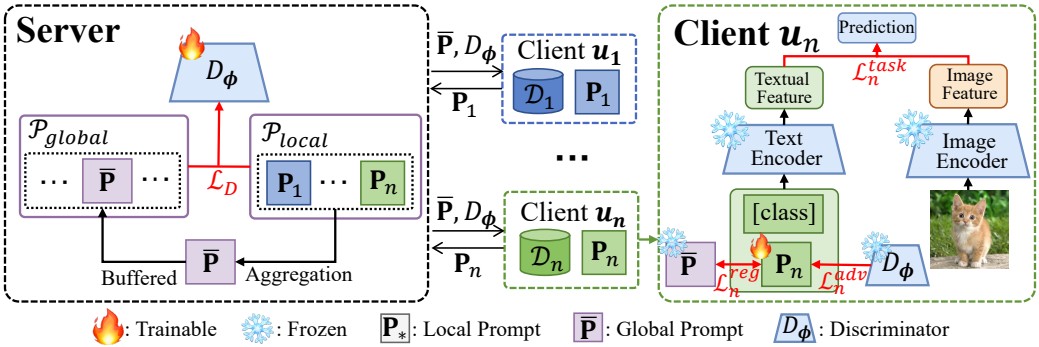

Figure 3: The server maintains a global prompt pool $\mathcal{P}_{global}$ and periodically aggregates client-uploaded local prompts $\{\mathbf{P}_1, \cdots, \mathbf{P}_n\}$ to obtain global prompt $\overline{\mathbf{P}}$. A discriminator $D_\phi$ is concurrently trained to distinguish between the distribution of local prompts and that of global prompts. During local training, each client $u_n$ receives the latest global prompt $\overline{\mathbf{P}}$ and the discriminator $D_\phi$ from server. $u_n$ then updates its prompt with guidance from: the local task loss ($\mathcal{L}_n^{task}$), the adversarial loss derived from $D_\phi$ ($\mathcal{L}_n^{adv}$), and the proximity regularization term from $\overline{\mathbf{P}}$ ($\mathcal{L}_n^{reg}$).

pacities. To address this challenge, model-heterogeneous federated learning (Alam et al., 2022; Ye et al., 2023; Jiang et al., 2024) has been introduced, allowing clients to adopt diverse model architectures. Knowledge distillation (KD) is a widely used strategy in this setting. A common strategy involves clients computing logits from their local models on a public dataset and uploading them to a server, where the logits are aggregated into global logits and redistributed (Li & Wang, 2019; Huang et al., 2022). Clients then refine their models by aligning local logits with global ones. However, this strategy is limited by the availability of a suitable public dataset and is prone to performance degradation when private data distributions diverge (Zhang et al., 2023). To avoid reliance on the public dataset, another line of work replaces the public dataset with a collaboratively trained auxiliary model, which is trained and shared among clients and used for mutual distillation to enhance their main models (Shen et al., 2020; Wu et al., 2022; Qin et al., 2023). While offering a viable solution to model heterogeneity, these methods introduce significant overheads associated with the training and dissemination of auxiliary models, particularly when applied to large-scale multi-modal models.

Beyond KD, prototype-based methods, such as FedProto (Tan et al., 2022) and FedTGP (Zhang et al., 2024a), propose exchanging label prototypes instead of model parameters, thereby allowing model heterogeneity. More recently, FedHPL (Ma et al., 2024) studies heterogeneity in a setting where clients share the same backbone family but with varying dimensions, incorporating prompt tuning to reduce computational overhead and using collaborative logit distillation while avoiding prompt aggregation. In contrast, we explore a fundamentally different and more challenging scenario: clients hold heterogeneous VLMs. Our focus is on resolving semantic misalignment across client-specific prompts, an aspect overlooked by aforementioned approaches.

## 2.2 FEDERATED PROMPT LEARNING

Recent advances in large-scale vision-language models (VLMs) (Radford et al., 2021; Zhang et al., 2024b) have shown strong generalization across various downstream tasks, making them conceptually aligned with the goals of FL. Nonetheless, deploying such models in FL settings remains non-trivial due to the considerable computational demands and high communication costs involved in training them in a distributed manner. Prompt learning (PL) (Zhou et al., 2022b; Liu et al., 2023) has become a flexible and efficient strategy for tailoring pre-trained models to specific tasks, utilizing learnable soft prompts in place of manually crafted ones. For example, CoOp (Zhou et al., 2022b) improves CLIP by substituting its static text templates with learnable prompts, and CoCoOp (Zhou et al., 2022a) builds upon this idea by conditioning prompt learning on image features, thereby further enhancing generalization and performance.

Building upon the efficiency of PL, PromptFL (Guo et al., 2024) presents the federated prompt learning (FPL) framework, which combines PL with the FedAvg (McMahan et al., 2017) to learn a

shared set of prompt vectors across distributed clients. Recent advances in FPL have increasingly focused on addressing statistical heterogeneity across clients. Notably, pFedPG (Yang et al., 2023) introduces a client-specific prompt generator on the server, facilitating the creation of personalized prompts for each client. pFedprompt (Guo et al., 2023) employs a non-parametric personalized attention module for each client, which generates localized visual features that are then integrated with the global textual prompt for prediction. FedOTP (Li et al., 2024) applies unbalanced Optimal Transport to improve the cooperation between global and local prompts. Meanwhile, FedPGP (Cui et al., 2024) utilizes pre-trained CLIP to guide the optimization of global prompts, enhancing their generalization, and incorporates a low-rank adaptation term to personalize them. Recently, DP-FPL (Tran et al., 2025) applies differential privacy to prompts, ensuring stronger and more rigorous privacy protection.

Although these FPL methods successfully integrate VLM into FL systems and deliver impressive performance, they are limited by the assumption that all clients share an *identical model architecture* for collaborative prompt learning. Such limitations restrict the applicability of existing methods in realistic federated settings, where clients typically maintain heterogeneous models due to hardware constraints, pre-training differences, or task-specific requirements. This model heterogeneity gives rise to significant semantic divergence in learned prompts, undermining performance and stability of collaborative learning. To fill this gap, we propose FedAPPR, a framework that systematically enforces semantic consistency across client prompts through adversarial alignment at the server and proximity regularization at the client to boost the collaborative training.

## 3 METHODOLOGY

### 3.1 FEDERATED PROMPT LEARNING

Prompt learning (PL) offers a parameter-efficient strategy to adapt pre-trained large models like CLIP (Radford et al., 2021) and ALIGN (Jia et al., 2021), by optimizing only a small number of prompt-specific parameters for downstream tasks. Unlike zero-shot CLIP, which uses a fixed word embedding matrix $\mathbf{W} = [\mathbf{w}_1, \mathbf{w}_2, \cdots, \mathbf{w}_e] \in \mathbb{R}^{d \times e}$ and relies on hand-crafted prompts such as "a photo of a [class]", with $e$ denoting the number of word embeddings and $d$ representing their dimensionality, PL replaces manual templates with $m$ trainable context vectors $\mathbf{P} = [\mathbf{p}_1, \mathbf{p}_2, \cdots, \mathbf{p}_m] \in \mathbb{R}^{d \times m}$, allowing the prompt itself to be learned directly from data. Accordingly, the prompt for class $i$ is constructed as $\mathcal{P}_i = \{\mathbf{w}_1, \mathbf{p}_1, \cdots, \mathbf{p}_m, \mathbf{w}_{m+2}, \cdots, \mathbf{w}_e\}$, where the original static embeddings $[\mathbf{w}_2, \cdots, \mathbf{w}_{m+1}]$ are substituted with learnable prompt vectors $[\mathbf{p}_1, \cdots, \mathbf{p}_m]$ to enhance adaptability in line with prior studies (Li et al., 2024; Cui et al., 2024). With both the text encoder $h(\cdot)$ and image encoder $g(\cdot)$ kept frozen, the likelihood of assigning an image $\mathbf{x}$ to class $i$ is determined by computing the similarity between the image embedding $g(\mathbf{x})$ and the prompt representation $h(\mathcal{P}_i)$:

$$q(\hat{y} = i \mid \mathbf{x}) = \frac{\exp\left(\text{sim}\left(g(\mathbf{x}), h\left(\mathcal{P}_i\right)\right)/t\right)}{\sum_{j=1}^{c} \exp(\text{sim}(g(\mathbf{x}), h(\mathcal{P}_j))/t)} \tag{1}$$

where $\text{sim}(\cdot, \cdot)$ denotes a similarity metric (e.g., cosine similarity), $\hat{y}$ is the predicted label, $c$ indicates the number of classes, and $t$ is a temperature hyperparameter. Given the training dataset $\mathcal{D}$, the learnable prompt parameters $\mathbf{P}$ can be optimized by minimizing the cross-entropy loss:

$$\mathcal{L}^{task} = -\frac{1}{|\mathcal{D}|} \sum_{(\mathbf{x}, \mathbf{y}) \in \mathcal{D}} \sum_{i=1}^{c} \mathbf{y}_i \log p(\hat{y} = i | \mathbf{x}) \tag{2}$$

where $\mathbf{y}$ denotes the one-hot encoded vector corresponding to the ground-truth label.

A straightforward approach for integrating PL into FL is to allow each client to locally optimize its prompt and intermittently upload them to a central server for aggregation and subsequent re-distribution. Formally, consider an FL system composed of $n$ clients $\{u_i\}_{i=1}^n$, where each client $u_i$ possesses a local dataset $\mathcal{D}_i = \{(\mathbf{x}_1, \mathbf{y}_1), (\mathbf{x}_2, \mathbf{y}_2), \cdots, (\mathbf{x}_{n_i}, \mathbf{y}_{n_i})\}$, which is strictly private and inaccessible to the central server or any other peers, and is equipped with a pre-trained VLM and a set of learnable prompt vectors $\mathbf{P}_i = [\mathbf{p}_{i,1}, \cdots, \mathbf{p}_{i,m}] \in \mathbb{R}^{d \times m}$. At each communication round, client $u_i$ updates its local prompt parameters $\mathbf{P}_i$ by minimizing the loss over its dataset using stochastic gradient descent:

$$\mathbf{P}_i = \mathbf{P}_i - \mu \nabla \mathcal{L}_i^{task}(\mathbf{P}_i; \mathcal{D}_i) \tag{3}$$

where $\mathcal{L}_i^{task}(\mathbf{P}_i; \mathcal{D}_i)$ represents the $u_i$'s local loss of $\mathbf{P}_i$ on dataset $\mathcal{D}_i$, and $\mu$ denotes the learning rate. After local training, clients transmit their refined prompts $\mathbf{P}_i$ to server to perform global averaging:

$$\overline{\mathbf{P}} = \sum\nolimits_{i=1}^{n} \frac{|\mathcal{D}_i|}{\sum_{j=1}^{n} |\mathcal{D}_j|} \mathbf{P}_i \tag{4}$$

where $\overline{\mathbf{P}}$ represents the aggregated prompt, which incorporates the knowledge accumulated from local clients. The overall optimization objective for FPL can thus be expressed as:

$$\overline{\mathbf{P}}^* = \arg\min_{\overline{\mathbf{P}}} \sum\nolimits_{i=1}^{n} \frac{|\mathcal{D}_i|}{\sum_{j=1}^{n} |\mathcal{D}_j|} \mathcal{L}_i^{task}(\overline{\mathbf{P}}; \mathcal{D}_i) \tag{5}$$

### 3.2 MODEL-HETEROGENEOUS FEDERATED PROMPT LEARNING

In the context of model-heterogeneous federated prompt learning (MHFPL), clients are equipped with different visual-language models (VLMs) due to varying computational capabilities or operating environments. As these models differ in architecture, they produce highly inconsistent prompt embeddings, which hinders effective prompt aggregation and thereby limits the model's performance, as illustrated in Figure 1. To address this challenge, we propose `FedAPPR`, a model-agnostic method designed for MHFPL, which enforces cross-model prompt alignment while preserving local adaptability via two complementary components: adversarial prompt alignment at the server level to align prompts across clients, and proximity regularization at the client level to stabilize prompt updates.

### 3.3 ADVERSARIAL PROMPT ALIGNMENT

In MHFPL, prompt embeddings optimized on non-identical VLMs often reside in disparate subspaces of the embedding space. Even if they encode similar semantics, the lack of architectural alignment results in representational drift across clients. To reconcile these divergences, we introduce an adversarial alignment mechanism based on a server-side discriminator that identifies whether prompts originate from a unified global distribution.

#### 3.3.1 DISCRIMINATOR TRAINING OBJECTIVE (SERVER)

At the beginning of each communication round $t$, the server maintains a historical buffer $\mathcal{P}_{global}$, containing prompts aggregated from previous rounds. These prompts have undergone semantic alignment through past global aggregation and thus serve as reliable references that represent the desired target distribution of aligned prompts. Simultaneously, each client uploads its locally updated prompt $\mathbf{P}_i$ to the server. Due to model and data heterogeneity across clients, these local prompts often differ from the globally aligned prompt space. Instead of discarding these variations, we leverage them as informative signals to guide alignment. Specifically, in each communication round, we construct a set $\mathcal{P}_{local}$ from the client-specific prompts, which serves as a reference for server-level adversarial alignment. The server then trains a binary discriminator $D_\phi(\cdot)$ that learns to distinguish between $\mathcal{P}_{global}$ and $\mathcal{P}_{local}$. The training objective is given by:

$$\mathcal{L}_D = -\mathbb{E}_{\mathbf{P} \sim \mathcal{P}_{global}}[\log D_\phi(\mathbf{P})] - \mathbb{E}_{\mathbf{P} \sim \mathcal{P}_{local}}[\log(1 - D_\phi(\mathbf{P}))]. \tag{6}$$

This objective encourages the discriminator to regard historical global prompts as aligned and to flag current local prompts as misaligned. Through this training process, the discriminator effectively constructs a semantic decision boundary that delineates aligned prompts from misaligned ones. As a result, $D_\phi(\cdot)$ provides a learnable reference frame that can guide the semantic calibration at client when tuning prompts in subsequent FL process. Together with the globally aggregated prompt, it is broadcasted to all clients to guide the next round of local training.

#### 3.3.2 CLIENT ADVERSARIAL OBJECTIVE

After receiving the current global prompt and discriminator, each client is instructed to refine its local prompt in an adversarial manner. Specifically, the discriminator $D_\phi(\cdot)$ serves as a semantic evaluator. The objective of each client is to adjust its prompt $\mathbf{P}_i$ such that it can "fool" the discriminator, making it indistinguishable from those in the global buffer. This adversarial alignment is achieved by minimizing the following loss:

$$\mathcal{L}_i^{adv} = -\log D_\phi(\mathbf{P}_i) \tag{7}$$

This adversarial objective encourages the local prompt $\mathbf{P}_i$ to be assigned with a higher confidence score by the discriminator, effectively pushing it toward the semantic region defined by $\mathcal{P}_{global}$. In doing so, the client is implicitly guided to align its prompt representation with the global prompt distribution, even under heterogeneous local model architectures.

### 3.3.3 WARM-UP STRATEGY

While adversarial alignment is critical for promoting semantic consistency among heterogeneous clients, applying it from the very beginning of training can be counterproductive. Specifically, early communication rounds suffer from two key challenges: (a) the global prompt $\mathcal{P}_{global}$ lacks semantic stability due to insufficient aggregation, and (b) the initial local prompts $\mathbf{P}_i$ are highly variable, reflecting random initialization or strong local biases. Under these conditions, directly optimizing local prompts to fool an immature discriminator may lead to unstable training dynamics, degraded task performance, and misaligned semantic representations.

To mitigate these risks, we introduce a warm-up strategy that defers the activation of adversarial training and global prompt buffer until the model has reached a more stable state. Concretely, during the first $T_0$ communication rounds (referred to as warm-up rounds), clients are trained solely with the task-specific loss (e.g., classification) and a proximity regularization term that will be introduced later. Once the warm-up period concludes (i.e., when $t > T_0$), the adversarial alignment objective is activated and integrated into the client optimization process. At this point, the global prompt provides a more reliable semantic reference, and the discriminator $D_\phi(\cdot)$ can be trained more effectively to guide the semantic alignment of the client prompts. This phased training scheme enhances the robustness of the overall learning process and improves convergence stability in model-heterogeneous federated prompt learning.

### 3.4 PROXIMITY REGULARIZATION

Although adversarial alignment provides a global mechanism for guiding local prompts toward semantic consistency, it alone may be insufficient to ensure stable and progressive alignment, especially under severe heterogeneity. To complement this, we introduce a client-level proximity regularization strategy that reinforces global consistency through local temporal coherence.

Specifically, instead of enforcing alignment solely via external feedback from the server-side discriminator, each client locally constrains its prompt updates across communication rounds. This is achieved by introducing a proximity regularization term that penalizes deviations from the reference prompt $\overline{\mathbf{P}}$, which corresponds to the global prompt received at the beginning of the current round:

$$\mathcal{L}_i^{reg} = ||\mathbf{P}_i - \overline{\mathbf{P}}||_2^2 \tag{8}$$

This regularization serves as an implicit consistency constraint: by anchoring the updated prompt $\mathbf{P}_i$ to the global reference $\overline{\mathbf{P}}$, it locally preserves the semantic direction induced by prior rounds of global aggregation. In this way, the proximity regularization steers each client toward the global prompt space, but does not override client-specific knowledge, which is maintained through the task-specific loss $\mathcal{L}_i^{task}$. By combining $\mathcal{L}_i^{reg}$ and $\mathcal{L}_i^{task}$, the framework balances global consistency with local personalization, enabling effective and stable collaborative learning.

### 3.5 OVERALL OPTIMIZATION OBJECTIVE

During each communication round, each client jointly updates its prompt parameters by minimizing a composite loss that integrates task-specific learning objectives with both global alignment and local consistency constraints. The overall optimization objective for client $u_i$ is defined as:

$$\mathcal{L}_i = \mathcal{L}_i^{task} + \lambda_a \mathcal{L}_i^{adv} + \lambda_r \mathcal{L}_i^{reg} \tag{9}$$

where $\mathcal{L}_i^{task}$ is the local supervised loss, $\mathcal{L}_i^{adv}$ is the adversarial alignment loss, $\mathcal{L}_i^{reg}$ is the proximity regularization loss, $\lambda_a$ and $\lambda_r$ are trade-off hyperparameters controlling the influence of the respective component.

After the optimization of local prompts, each client uploads its prompts to the server. Based on the aggregation strategy defined in Eq. (4), the server agglomerates these local prompts into a global prompt. This global prompt serves as the semantic anchor for subsequent client updates, enabling

consistent knowledge transfer across rounds. Algorithm S1 in Appendix summarizes the pseudo-code of `FedAPPR`, which comprises prompt update on clients (Steps 4-8), and prompt aggregation and discriminator training on the server (Steps 11-14). Discussion on communication and computational costs are provided in Section C of Appendix.

## 4 EXPERIMENTS

### 4.1 EXPERIMENTAL SETUP

**Datasets.** To comprehensively evaluate the effectiveness of `FedAPPR`, we adopt six widely recognized datasets commonly used in CLIP-related research (Radford et al., 2021). These datasets span multiple categories of tasks: General object recognition: Caltech101 (Fei-Fei et al., 2004), CIFAR-10 (Krizhevsky et al., 2009); Fine-grained visual classification: CIFAR-100 (Krizhevsky et al., 2009), Flowers-102 (Nilsback & Zisserman, 2008), Oxford Pets (Parkhi et al., 2012); Action recognition: UCF101 (Soomro, 2012). For each dataset, we simulate two types of data distribution across clients: i.i.d. and non-i.i.d. In the i.i.d. setting, data samples are uniformly and randomly assigned to clients. In the non-i.i.d. scenario, we follow a widely adopted practice by employing a Dirichlet distribution to partition the data, where the degree of heterogeneity is controlled by a Dirichlet parameter $\alpha$, which also leads to different numbers of samples across clients. A lower value of $\alpha$ indicates a more skewed distribution, resulting in greater heterogeneity among client data. In main experiments, we use $\alpha = 0.1$ as the default setting for heterogeneous partitioning (Lin et al., 2020; Zhang et al., 2024a). We also conduct additional analyses on varying $\alpha$ in the Appendix.

**Baselines.** We conduct experiments using two types of baselines: (1) Local: (i) **Zero-shot CLIP (CLIP-ZS)** (Radford et al., 2021), which uses manually crafted text prompt templates (e.g., "a photo of a [class]"), and (ii) **CoOp (Local Train)** (Zhou et al., 2022b), where each client independently learns prompt vectors using local data. (2) Federated Learning: (i) FedProto (Tan et al., 2022), (ii) FedHPL (Ma et al., 2024), (iii) PromptFL (Guo et al., 2024), (iv) pFedPrompt (Guo et al., 2023), (v) FedOTP (Li et al., 2024), and (vi) FedPGP (Cui et al., 2024). For all baseline methods, we report the mean and standard deviation of test accuracy over five independent trials. In each trial, we calculate the average test accuracy across all clients.

**Heterogeneous models.** To simulate scenarios with model heterogeneity, we assign each client a backbone randomly selected from a set of four widely used architectures in CLIP: ResNet-50 (He et al., 2016), ResNet-101 (He et al., 2016), ViT-B/32 (Dosovitskiy et al., 2021), and ViT-B/16 (Dosovitskiy et al., 2021).

**Implementation Details.** To ensure a fair comparison, all methods are configured with the following settings: CLIP as the local model; the SGD optimizer with learning rates of 0.001; the batchsize of 64; 1 epoch in each local update; 100 communication rounds; 60%/20%/20% partition of the local data for the training/validation/testing dataset; cosine similarity as the metric function; $n = 20$ clients with 100% participation or $n = 100$ clients with 20% participation; the length of prompt vectors $m = 16$ with a dimension $d = 512$. As to the hyper-parameter of our `FedAPPR`, the discriminator is configured as the two-layer fully connected layer with ReLU non-linearity, the trade-off parameter $\lambda_a = 0.1$, $\lambda_r = 0.01$ and Warm-up round $T_0 = 20$.

More details about the datasets, baselines, and implementation specifics are provided in Appendix.

### 4.2 EXPERIMENTAL RESULTS

We conduct experiments on six datasets across two data distribution scenarios with 20 or 100 clients and report the corresponding results in Table 1. Due to page limitations, the results for scenarios with 20 clients are provided in the Appendix. Observations derived from these tables include:

(i) `FedAPPR` almost achieves superior performance across the six datasets and under both i.i.d. and non-i.i.d. data distributions. This demonstrates the strong generalization ability and robustness of our approach in diverse scenarios. When scaling the number of clients from 20 to 100, most methods experience performance degradation due to the reduced training data per client caused by increased data partitioning. CLIP-ZS, however, remains unaffected, as it performs zero-shot inference without relying on client-side training. On the other hand, transitioning from i.i.d. to non-i.i.d. data distributions leads to a noticeable accuracy drop for most FL methods, primarily due to the heightened statistical heterogeneity among clients, which complicates the learning of effective prompts.

Table 1: Accuracy (mean%±std%) of comparison methods on six datasets under scenarios with 100 clients. The best performance in each setting is **bold-faced**.

| Dataset | Caltech101 | | CIFAR-10 | | CIFAR-100 | |
|---|---|---|---|---|---|---|
| Distribution | i.i.d. | non-i.i.d. | i.i.d. | non-i.i.d. | i.i.d. | non-i.i.d. |
| CLIP-ZS | 75.67±0.62 | 75.69±0.71 | 80.23±0.63 | 80.12±0.69 | 57.58±0.65 | 56.99±0.73 |
| Local Train | 85.43±0.63 | 85.03±0.59 | 89.56±0.43 | 89.59±0.60 | 73.34±0.65 | 73.01±0.56 |
| FedProto | 83.09±0.48 | 82.55±0.33 | 87.18±0.45 | 86.49±0.37 | 71.35±0.59 | 70.72±0.78 |
| FedHPL | 85.42±0.50 | 84.69±0.44 | 89.86±0.55 | 89.08±0.67 | 74.39±0.62 | 73.60±0.78 |
| PromptFL | 84.39±0.67 | 83.28±0.42 | 88.53±0.55 | 87.53±0.53 | 72.10±0.69 | 71.20±0.62 |
| pFedPrompt | 85.30±0.63 | 84.56±0.60 | 89.63±0.49 | 88.52±0.40 | 73.69±0.50 | 72.72±0.65 |
| FedOTP | 85.39±0.46 | 84.38±0.63 | 90.20±0.50 | 89.33±0.38 | 74.06±0.26 | 73.19±0.36 |
| FedPGP | 85.53±0.40 | 84.63±0.51 | 90.47±0.58 | 89.65±0.43 | 73.95±0.38 | 73.02±0.44 |
| FedAPPR | **87.65±0.39** | **86.56±0.43** | **92.53±0.43** | **91.53±0.50** | **75.85±0.39** | **75.03±0.49** |
| Dataset | Flower102 | | OxfordPets | | UCF101 | |
| Distribution | i.i.d. | non-i.i.d. | i.i.d. | non-i.i.d. | i.i.d. | non-i.i.d. |
| CLIP-ZS | 53.39±0.56 | 53.06±0.82 | 74.07±0.73 | 73.89±0.60 | 51.03±0.56 | 51.25±0.53 |
| Local Train | 79.83±0.49 | 79.92±0.55 | 81.17±0.53 | 80.86±0.49 | 74.26±0.49 | 74.06±0.59 |
| FedProto | 77.39±0.43 | 76.89±0.55 | 79.56±0.47 | 78.82±0.56 | 72.90±0.52 | 72.07±0.61 |
| FedHPL | 80.56±0.60 | 79.88±0.52 | 81.29±0.42 | 80.50±0.60 | 74.09±0.66 | 73.72±0.52 |
| PromptFL | 79.29±0.53 | 78.26±0.47 | 80.09±0.59 | 79.12±0.73 | 73.41±0.46 | 72.65±0.73 |
| pFedPrompt | 80.55±0.43 | 79.61±0.55 | 80.86±0.44 | 80.02±0.63 | 74.16±0.52 | 73.25±0.60 |
| FedOTP | 80.93±0.30 | 80.12±0.59 | 81.73±0.52 | 80.83±0.39 | 74.79±0.59 | 73.83±0.53 |
| FedPGP | 80.76±0.45 | 80.09±0.40 | 81.99±0.49 | 80.76±0.52 | **75.78±0.33** | 74.81±0.39 |
| FedAPPR | **82.86±0.36** | **81.77±0.59** | **83.26±0.40** | **82.39±0.37** | 75.59±0.58 | **75.05±0.39** |

(ii) **Local** vs. **FL**: While LIP-ZS exhibits strong zero-shot generalization, its inability to adapt to specific tasks constrains its performance. Local Train addresses this limitation by incorporating learnable prompts tailored to the target tasks. PromptFL often lags behind Local Train due to large prompt divergence caused by model heterogeneity, which even makes the global prompt hard to converge to a universally effective representation across diverse client models. However, Local Train are consistently outperformed by our FedAPPR, which enables meaningful collaboration among clients through prompt alignment that mitigates prompt divergence across heterogeneous clients.

(iii) FedAPPR **vs. FedProto and FedHPL**: FedAPPR consistently outperforms both FedProto and FedHPL across settings, revealing that conventional prototype- or distillation-based approaches ignore the semantic divergence of prompt spaces, which consequently hampers their effectiveness to handle heterogeneous models.

(vi) FedAPPR **vs. PromptFL, pFedPrompt, FedOTP and FedPGP**: Although pFedPrompt, FedOTP and FedPGP adopt prompt personalization to achieve better performance than PromptFL, they still struggle to surpass Local Train. This is due to their limited ability to coordinate personalization with cross-client model-heterogeneity. In contrast, our FedAPPR explicitly addresses prompt divergence by integrating prompt alignment that balances local adaptability with shared semantic consistency, enabling more effective and stable learning across heterogeneous clients.

## 4.3 FURTHER ANALYSIS

**Ablation Studies.** We conduct an ablation study to investigate the individual contributions of key components in FedAPPR. To this end, we design three ablated variants of FedAPPR: FedAPPR-nA excludes the adversarial loss used for prompt alignment; FedAPPR-nW removes the warm-up strategy, which is introduced to stabilize the training process; FedAPPR-nR eliminates the proximity regularization, which enforces further consistency constraints. Table 2 presents results of three variants and FedAPPR on Caltech101, CIFAR-10 and CIFAR-100 under 100 clients, while results on Flower102, OxfordPets and UCF101 are reported in the Appendix. From the tables, we observe that: (i) Removing any component leads to a consistent drop in performance across datasets and data distributions, validating the importance of each design choice. (ii) The absence of the adversarial loss (FedAPPR-nA) results in the most significant degradation, highlighting the crucial contribution of prompt alignment to the overall effectiveness of the method. (iii) Excluding the warm-up strategy (FedAPPR-nW) reduces performance, indicating its stabilizing effect during discriminator training. (iv) Disabling the proximity regularization (FedAPPR-nR) also yields performance decline, highlighting its contribution to model consistency across clients.

Table 2: Accuracy (mean%±std%) of `FedAPPR` and its variants under scenarios with 100 clients. The best performance in each setting is **bold-faced**.

| Dataset | Caltech101 | | CIFAR-10 | | CIFAR-100 | |
|---|---|---|---|---|---|---|
| Distribution | i.i.d. | non-i.i.d. | i.i.d. | non-i.i.d. | i.i.d. | non-i.i.d. |
| `FedAPPR-nA` | 85.39±0.48 | 84.39±0.52 | 90.35±0.43 | 89.83±0.49 | 73.64±0.36 | 73.18±0.39 |
| `FedAPPR-nW` | 86.03±0.40 | 84.93±0.53 | 90.89±0.50 | 90.22±0.35 | 73.99±0.49 | 73.53±0.52 |
| `FedAPPR-nR` | 86.33±0.39 | 85.41±0.30 | 91.27±0.50 | 90.39±0.56 | 73.83±0.51 | 73.92±0.66 |
| `FedAPPR` | **87.65±0.39** | **86.56±0.43** | **92.53±0.43** | **91.53±0.50** | **75.85±0.39** | **75.03±0.49** |

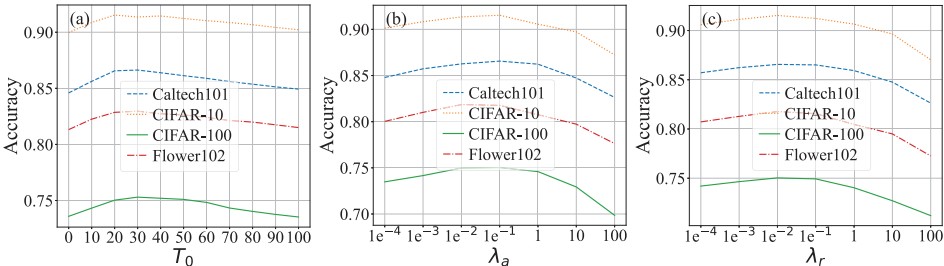

Figure 4: (a) Accuracy of `FedAPPR` vs. Warm-up Rounds ($T_0$); (b) Accuracy of `FedAPPR` vs. $\lambda_a$; (c) Accuracy of `FedAPPR` vs. $\lambda_r$. Both are conducted under non-i.i.d. scenarios with 100 clients.

**Study on Warm-up Strategy.** In this experiment, we investigate the effect of the warm-up strategy on performance by varying the number of warm-up rounds $T_0$ from 0 to 100 under non-i.i.d. scenarios with 100 clients. As shown in Figure 4 (a), the accuracy of `FedAPPR` initially increases with $T_0$, reaching a peak around $T_0 = 20 \sim 30$, and then gradually declines as $T_0$ continues to increase. This trend highlights the trade-off between premature adversarial activation and delayed alignment. When $T_0$ is too small, adversarial alignment is introduced prematurely, while the global prompt $\mathcal{P}_{\text{global}}$ and the local prompts $\mathbf{P}_i$ are still unstable. This results in noisy gradient signals from the discriminator and undermines the alignment process, leading to degraded performance. On the other hand, setting $T_0$ too large delays the introduction of adversarial supervision, which limits the benefits of semantic alignment in later training stages.

**Effect of Trade-off Parameters.** To evaluate the influence of trade-off parameters on model performance, we conduct sensitivity analyses on the adversarial weight $\lambda_a$ and the regularization weight $\lambda_r$, both of which balance different components of the local training objective in `FedAPPR`, by varying it within the range of $\{0.0001, 0.001, \cdots, 10, 100\}$. The results under non-i.i.d. scenarios with 100 clients are shown in Figure 4 (b) and (c). (i) For $\lambda_a$, accuracy steadily improves as the value increases up to approximately $\lambda_a \approx 0.1$, highlighting the benefits of stronger adversarial alignment. However, as $\lambda_a$ continues to increase, accuracy drops significantly due to the adversarial objective overpowering the primary task loss, leading to unstable updates and degraded performance. (ii) For $\lambda_r$, accuracy similarly improves with increasing $\lambda_r$ up to around $\lambda_r \approx 0.01$, demonstrating the advantage of moderate proximity regularization. This guides local prompts toward the global semantic space. Beyond this range, however, excessive regularization restricts the flexibility of local prompts, resulting in underfitting and reduced performance.

Further experimental results, including convergence analysis, robustness to data heterogeneity, large-scale evaluations, and complexity analysis, are provided in Appendix.

## 5 CONCLUSION

In this paper, we focus on an important and practical but unexplored FPL problem: model-heterogeneous federated prompt learning (MHFPL), where clients adopt different model architectures due to their varying computational resources. To tackle the significant prompt divergence challenges arising from model heterogeneity, our `FedAPPR` introduces the server-level adversarial alignment and client-level proximity regularization to ensure semantic consistency and reduce prompt variance across clients. Extensive experiments on six benchmark datasets demonstrate the superiority of `FedAPPR`, highlighting its effectiveness in handling heterogeneous VLMs scenarios and affirming it as a promising approach for FL with large-scale VLMs. We hope that this work paves the way for future research in FPL in more realistic and heterogeneous environments.

## 6 ETHICS STATEMENT

We promise that we have read the ICLR Code of Ethics, and this article has not raised any questions regarding the Code of Ethics.

## 7 REPRODUCIBILITY STATEMENT

We have made extensive efforts to ensure the reproducibility of our results:
**Datasets**: All benchmark datasets are publicly available, and their sources are cited in the paper.
**Baselines**: We carefully reproduce the baselines and configure them with the recommended parameters reported in the respective literature to ensure fairness.
**Hyperparameters**: All hyperparameter configurations, training schedules, and optimization strategies are explicitly reported in the main text and Appendix.
**Implementation**: We provide the full implementation of `FedAPPR` in the main paper or Appendix, including model definitions, training procedures, evaluation scripts, and other essential components.
**Environment**: The software environment and hardware specifications are documented in the Appendix.

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

# Appendix

## A  THE USE OF LARGE LANGUAGE MODELS

The use of large language models (LLMs) in this work is limited to language polishing and grammar improvement.

## B  ALGORITHM TABLE

---

**Algorithm S1** `FedAPPR`: **Fed**erated **A**dversarial **P**rompt Alignment and **P**roximity **R**egularization

---

**Input**: $n$ clients, where each client $u_i$ carries local data $\mathcal{D}_i$ and learnable prompts $\mathbf{P}_i$; $T = 100$ (number of communication rounds).

**Output**: trained personalized prompts $\{\mathbf{P}_1^*, \cdots, \mathbf{P}_n^*\}$.

 1: *Server* initializes clients' prompt using $\mathbf{P}$.
 2: **for** $t = 1 \rightarrow T$ **do**
 3:    **for** all *clients* $i = 1 \rightarrow n$ **in parallel do**
 4:        Receive aggregated prompts $\overline{\mathbf{P}}$ and discriminator $D_\phi$ from server.
 5:        Compute the adversarial alignment loss $\mathcal{L}_i^{adv}$ as Eq. (7).
 6:        Calculate the proximity regularization loss $\mathcal{L}_i^{reg}$ as Eq. (8).
 7:        Compute the overall local loss as in Eq. (9) and update the local prompts accordingly.
 8:        Upload updated prompts $\mathbf{P}_i$ to server.
 9:    **end for**
10:    **for** *server* **do**
11:        Receive local prompts $\{\mathbf{P}_i\}_{i=1}^n$ from $n$ clients.
12:        Aggregate local prompts using Eq. (4) to obtain global one and buffer it into global pool.
13:        Calculate discriminator training loss using Eq. (6) and optimize the discriminator accordingly.
14:        Broadcast global prompts $\overline{\mathbf{P}}$ and discriminator $D_\phi$ to respective clients.
15:    **end for**
16: **end for**

---

Algorithm S1 summarizes the pseudo-code of `FedAPPR`, which comprises prompt update on clients (Steps 4-8), and prompt aggregation and discriminator training on the server (Steps 11-14).

## C  DISCUSSION ABOUT COMMUNICATION AND COMPUTATIONAL COST

**Communication Cost.** `FedAPPR` only exchanges prompt vectors and a lightweight discriminator between clients and the server in each round. Therefore, the additional communication overhead compared to PromptFL is $CL(D_\phi)$, where $CL(*)$ denotes the communication load of $*$. Given that the discriminator consists of only two fully connected layers (as detailed in Section 4), this added communication cost remains minimal and acceptable.

**Computational Overhead.** On the client side, the additional computational cost mainly stems from the adversarial alignment loss and proximity regularization loss used to guide prompt training, which introduces only minimal overhead due to their lightweight computation. On the server side, the primary overhead arises from training the discriminator. As the discriminator in `FedAPPR` comprises only two fully connected layers, its training cost remains significantly lower than that of local training.

## D  EXPERIMENTAL SETUP

### D.1  DATASETS

To comprehensively evaluate the effectiveness of `FedAPPR`, we adopt six widely recognized datasets commonly used in CLIP-related research (Radford et al., 2021). These datasets span mul-

Table S1: Statistics of six datasets.

| Dataset | #Classes | #Samples | Task |
|---|---|---|---|
| Caltech101 | 101 | 6,593 | Object recognition |
| CIFAR-10 | 10 | 60,000 | Image Classification |
| CIFAR-100 | 100 | 60,000 | Fine-grained image classification |
| Flower102 | 102 | 6,556 | Fine-grained flower recognition |
| OxfordPets | 37 | 6,613 | Fine-grained pet recognition |
| UCF101 | 101 | 11,422 | Action Recognition |

tiple categories of tasks: General object recognition: Caltech101 (Fei-Fei et al., 2004), CIFAR-10 (Krizhevsky et al., 2009); Fine-grained visual classification: CIFAR-100 (Krizhevsky et al., 2009), Flowers-102 (Nilsback & Zisserman, 2008), Oxford Pets (Parkhi et al., 2012); Action recognition: UCF101 (Soomro, 2012). Detailed statistics for these datasets are provided in Table S1.

### D.2 BASELINES

We conduct experiments using two types of baselines: (1) Local: (i) **Zero-shot CLIP (CLIP-ZS)** (Radford et al., 2021), and (ii) **CoOp (Local Train)** (Zhou et al., 2022b). (2) Federated Learning: (i) FedProto (Tan et al., 2022), (ii) FedHPL (Ma et al., 2024), (iii) PromptFL (Guo et al., 2024), (iv) pFedPrompt (Guo et al., 2023), (v) FedOTP (Li et al., 2024), and (vi) FedPGP (Cui et al., 2024). These methods are configured with the suggested parameters from the respective literature:

- **CLIP-ZS** uses manually crafted text prompt templates (e.g., "a photo of a [class]") to perform zero-shot prediction.
- **Local Train** refers to the strategy where each client individually optimizes prompt vectors using its own local dataset.
- **FedProto** shares local prototypes instead of models and executes prototype aggregation on the server by averaging the prototype sent from participating clients. Suggested configuration: the trade-off parameter $\lambda = 0.1$.
- **FedHPL** relies on global logit distillation with weighted aggregation to address model heterogeneity in federated learning, while using lightweight prompt tuning only for efficient local adaptation. Suggested configuration: the temperature parameter $\mathcal{T} = 4.5$ and the trade-off parameter $\gamma = 1$.
- **PromptFL** combines prompt learning with FedAvg (McMahan et al., 2017), enabling the learning of a unified set of prompt vectors from distributed datasets across multiple clients.
- **pFedPrompt** maintains a non-parametric personalized attention module for each client, which generates locally personalized visual features and combines them with the global textual prompt for prediction. Suggested configuration: trade-off parameter $\alpha = 0.5$.
- **FedOTP** employs unbalanced Optimal Transport to better align and coordinate global and local prompts for improved cooperative learning. Suggested configuration: regularization parameter $\lambda = 0.1$ and maximum iteration number $iter = 100$.
- **FedPGP** employs a pre-trained CLIP model to guide the optimization of globally shared prompts, thereby improving their generalization across clients, and further integrates a low-rank adaptation term to enable client-specific personalization. Suggested configuration: low-rank decomposition bottleneck $b = 8$, and the trade-off parameter $\mu = 1$.

### D.3 IMPLEMENTATION DETAILS

We implement all the methods using PyTorch 1.13 and conduct experiments on a server with the Intel(R) Xeon(R) Gold 6248R CPU, 512G memory, 2 NVIDIA L40 GPUs, and Ubuntu 22.04.

### D.4 IMPLEMENTATION OF FIGURE 1.

For the experiments illustrated in Figure 1, we perform evaluations on the Caltech101 dataset under an i.i.d. setting with 100 clients. In the model-homogeneous scenario, all clients utilize the ViT-B/16 backbone. In contrast, the model-heterogeneous scenario assigns each client a model randomly chosen from four commonly used architectures: ResNet-50, ResNet-101, ViT-B/32, and ViT-B/16. To visualize the learned prompts, we employ the t-SNE (Van der Maaten & Hinton, 2008).

Table S2: Accuracy (mean%±std%) of comparison methods on six datasets under scenarios with 20 clients. The best performance in each setting is **bold-faced**.

| Dataset | Caltech101 | | CIFAR-10 | | CIFAR-100 | |
|---|---|---|---|---|---|---|
| Distribution | i.i.d. | non-i.i.d. | i.i.d. | non-i.i.d. | i.i.d. | non-i.i.d. |
| CLIP-ZS | 75.56±0.53 | 75.93±0.62 | 80.01±0.62 | 80.62±0.55 | 57.01±0.76 | 56.53±0.69 |
| Local Train | 87.59±0.50 | 87.79±0.63 | 91.29±0.46 | 91.20±0.52 | 75.53±0.54 | 75.79±0.42 |
| FedProto | 85.98±0.47 | 85.19±0.56 | 89.85±0.63 | 89.09±0.50 | 73.67±0.64 | 72.89±0.50 |
| FedHPL | 88.20±0.39 | 87.39±0.59 | 91.26±0.59 | 90.58±0.49 | 75.92±0.48 | 75.05±0.62 |
| PromptFL | 86.13±0.60 | 85.26±0.39 | 90.25±0.50 | 89.03±0.53 | 74.35±0.26 | 73.33±0.53 |
| pFedPrompt | 87.79±0.48 | 87.03±0.52 | 91.59±0.56 | 90.52±0.40 | 75.93±0.45 | 75.19±0.50 |
| FedOTP | 88.28±0.39 | 87.65±0.57 | 91.89±0.47 | 91.23±0.38 | 75.86±0.39 | 75.17±0.41 |
| FedPGP | 88.52±0.40 | 88.11±0.52 | 91.29±0.62 | 91.09±0.54 | 76.03±0.46 | 75.12±0.61 |
| FedAPPR | **90.63±0.43** | **89.99±0.35** | **94.26±0.36** | **92.53±0.50** | **77.97±0.52** | **77.56±0.53** |
| Dataset | Flower102 | | OxfordPets | | UCF101 | |
| Distribution | i.i.d. | non-i.i.d. | i.i.d. | non-i.i.d. | i.i.d. | non-i.i.d. |
| CLIP-ZS | 53.55±0.60 | 53.29±0.66 | 74.35±0.60 | 74.11±0.36 | 51.50±0.49 | 51.44±0.39 |
| Local Train | 82.66±0.52 | 81.73±0.49 | 83.03±0.43 | 82.72±0.60 | 76.53±0.50 | 75.46±0.86 |
| FedProto | 80.83±0.47 | 80.10±0.59 | 81.59±0.57 | 80.89±0.40 | 74.88±0.52 | 74.02±0.60 |
| FedHPL | 82.29±0.42 | 81.58±0.67 | 82.97±0.50 | 82.13±0.45 | 76.61±0.43 | 75.98±0.58 |
| PromptFL | 81.53±0.60 | 80.67±0.35 | 82.23±0.43 | 81.65±0.62 | 75.61±0.39 | 74.88±0.62 |
| pFedPrompt | 82.56±0.49 | 81.53±0.63 | 83.13±0.62 | 82.77±0.39 | 77.07±0.62 | 75.69±0.56 |
| FedOTP | 83.02±0.42 | **83.50±0.62** | 83.38±0.68 | 82.68±0.47 | 76.79±0.48 | 76.01±0.40 |
| FedPGP | 82.86±0.35 | 82.19±0.55 | 83.50±0.62 | 82.87±0.56 | 76.53±0.52 | 75.88±0.53 |
| FedAPPR | **84.05±0.39** | 83.34±0.39 | **85.62±0.48** | **84.58±0.32** | **78.20±0.50** | **77.74±0.42** |

Table S3: Accuracy (mean%±std%) of FedAPPR and its variants under scenarios with 100 clients. The best performance in each setting is **bold-faced**.

| Dataset | Flower102 | | OxfordPets | | UCF101 | |
|---|---|---|---|---|---|---|
| Distribution | i.i.d. | non-i.i.d. | i.i.d. | non-i.i.d. | i.i.d. | non-i.i.d. |
| FedAPPR-nA | 81.23±0.39 | 79.73±0.35 | 81.03±0.48 | 80.06±0.49 | 73.59±0.42 | 73.00±0.39 |
| FedAPPR-nW | 81.50±0.43 | 80.13±0.47 | 81.47±0.49 | 80.59±0.55 | 74.03±0.39 | 73.18±0.40 |
| FedAPPR-nR | 81.79±0.40 | 80.42±0.39 | 81.89±0.65 | 80.83±0.42 | 74.05±0.56 | 73.43±0.43 |
| FedAPPR | **82.86±0.36** | **81.77±0.59** | **83.26±0.40** | **82.39±0.37** | **75.59±0.58** | **75.05±0.39** |

## D.5 Implementation of Figure 2.

For the experiments in Figure 2, we adopt PromptFL to conduct experiments on the Caltech101 dataset under an i.i.d. scenario with 100 clients and model heterogeneity. Clients are grouped into four sets of 25 based on different VLM backbones, and their individual accuracies are reported. Clear performance disparities across groups highlight the impact of model heterogeneity.

# E ADDITIONAL RESULTS

## E.1 ADDITIONAL RESULTS OF MAIN EXPERIMENTS

Table S2 presents the comparison results in scenarios involving 20 clients. The results show that the proposed FedAPPR consistently outperforms the baselines across various settings, and the conclusions are consistent with those of the main manuscript.

## E.2 ADDITIONAL RESULTS OF ABLATION STUDY

Table S2 presents the additional ablation results on Flower102, OxfordPets, and UCF101 under scenarios with 100 clients. The results again validate the effectiveness of each component in the proposed FedAPPR.

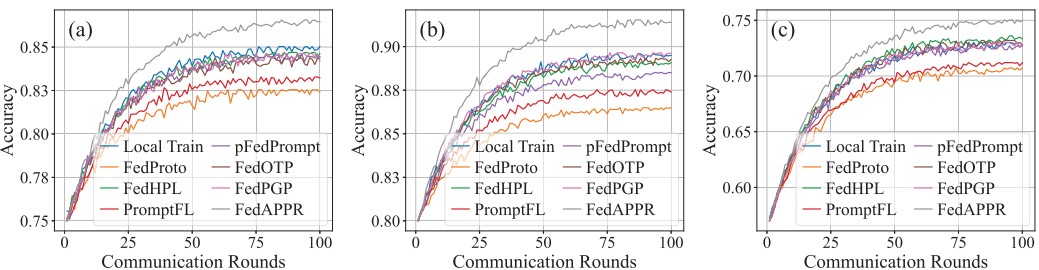

Figure S1: Accuracy of compared methods vs. communication round under non-i.i.d. scenarios with 100 clients on (a) Caltech101, (b) CIFAR-10 and (c) CIFAR-100.

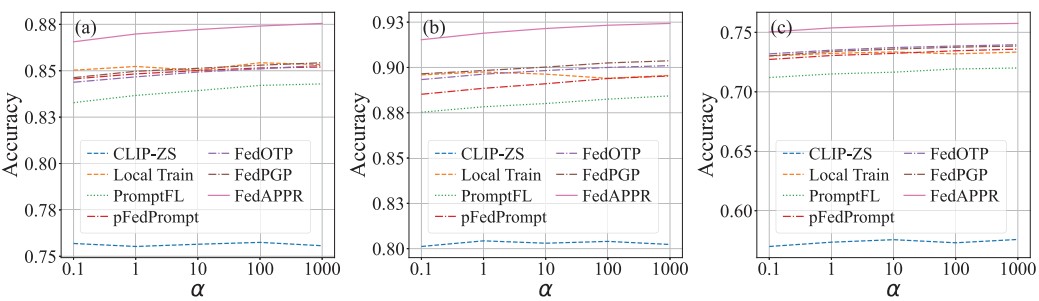

Figure S2: Accuracy of compared methods vs. $\alpha$ under 100-client scenarios on (a) Caltech101, (b) CIFAR-10 and (c) CIFAR-100.

### E.3 Convergence Analysis

Figure S1 depicts the test accuracy of compared methods throughout the training process on three datasets under non-i.i.d. scenarios with 100 clients. The figures reveal that `FedAPPR` attains its peak accuracy within approximately 75 communication rounds. This observation highlights a comparable convergence rate compared to other methods, accompanied by exceptional classification accuracy. These results once again emphasize the practicality and superiority of `FedAPPR` in addressing model heterogeneity among clients.

### E.4 Different Levels of Data Distribution Heterogeneity

To evaluate the robustness of compared methods under varying degrees of data heterogeneity, we simulate non-i.i.d. conditions by adjusting the Dirichlet distribution parameter $\alpha$ across the set $\{0.1, 1, 10, 100, 1000\}$. Smaller values of $\alpha$ result in highly skewed distributions, while larger values approximate uniform data allocation across clients. Experimental results on Caltech101, CIFAR-10, and CIFAR-100 (see Figure S2) reveal that model performance consistently improves as $\alpha$ increases, highlighting the positive correlation between data homogeneity and collaborative learning efficiency. Notably, our proposed `FedAPPR` demonstrates stable and superior accuracy across all levels of heterogeneity, underscoring its strong generalization capability in federated environments.

### E.5 Accuracy with larger scale clients

We conduct further experiments under a larger-scale FL setting with 1,000 clients and 10% participation in each communication round. The results in Table S4 show that `FedAPPR` consistently achieves the highest accuracy across both datasets and distributions, outperforming all baselines by a notable margin. This highlights `FedAPPR`'s potential for real-world deployment in practical FL scenarios with large client populations.

Table S4: Accuracy (mean%±std%) of compared methods under non-i.i.d. scenarios with 1000 clients.

| Dataset | CIFAR-10 | | CIFAR-100 | |
|---|---|---|---|---|
| Distribution | i.i.d. | non-i.i.d. | i.i.d. | non-i.i.d. |
| CLIP-ZS | 80.15±0.62 | 80.39±0.55 | 56.01±0.40 | 56.78±0.48 |
| Local Train | 86.58±0.45 | 85.63±0.75 | 70.67±0.43 | 70.29±0.74 |
| FedProto | 84.62±0.39 | 84.01±0.56 | 68.74±0.48 | 67.95±0.70 |
| FedHPL | 87.49±0.50 | 86.66±0.30 | 70.41±0.57 | 69.98±0.54 |
| PromptFL | 85.07±0.39 | 84.40±0.62 | 69.25±0.55 | 68.73±0.55 |
| pFedPrompt | 86.83±0.50 | 85.52±0.66 | 70.89±0.42 | 69.98±0.47 |
| FedOTP | 87.53±0.49 | 86.72±0.46 | 71.38±0.42 | 70.56±0.50 |
| FedPGP | 87.82±0.64 | 86.97±0.43 | 70.79±0.45 | 70.25±0.47 |
| FedAPPR | **89.77±0.39** | **88.94±0.38** | **72.73±0.41** | **72.26±0.58** |

E.6    COMMUNICATION AND COMPUTATIONAL OVERHEAD

To further address concerns about complexity, we now provide **quantitative comparisons** on: (i) **Communication cost per round** (in transmitted parameters), and (ii) **Computation time per round** (in average runtime). The results under non-i.i.d. scenarios with 100 clients are reported.

As shown in Table S5, FedAPPR achieves favorable computational efficiency, with runtime consistently lower than FedHPL, FedOTP and FedPGP, and competitive with FedProto, PromptFL and pFedPrompt.

Furthermore, Table S6 confirms that the communication overhead of FedAPPR is only marginally higher than that of other prompt-based methods, primarily due to the inclusion of a lightweight discriminator with $d_1 \times d_2 + d_2$ parameters. Specifically, with a prompt dimension of $d = 512$, prompt length $m = 16$, and discriminator dimensions $d_1 = 512$, $d_2 = 8$, the additional overhead introduced by the discriminator amounts to $512 \times 8 + 8 \times 1 = 4{,}104$ parameters. This is negligible compared to the prompt transmission size of $m \times d \times 2 = 16 \times 512 \times 2 = 16{,}384$ parameters. The cost is also comparable to that of FedProto and FedHPL, where $p = 512$ denotes the prototype dimension and $l$ the number of labels.

Table S5: Runtime (seconds) of compared methods within one communication round under non-i.i.d. scenarios with 100 clients.

| Dataset | Caltech101 | CIFAR-10 | CIFAR-100 | Flower102 | OxfordPets | UCF101 |
|---|---|---|---|---|---|---|
| FedProto | 46.5 | 166.8 | 190.1 | 69.9 | 53.6 | 84.5 |
| FedHPL | 65.8 | 238.5 | 299.1 | 109.2 | 85.3 | 139.2 |
| PromptFL | 39.7 | 147.2 | 169.7 | 62.7 | 47.6 | 78.5 |
| pFedPrompt | 52.6 | 214.6 | 236.2 | 88.2 | 63.7 | 94.2 |
| FedOTP | 63.4 | 241.3 | 285.0 | 104.6 | 80.0 | 134.5 |
| FedPGP | 68.6 | 258.6 | 318.8 | 116.2 | 89.2 | 147.6 |
| FedAPPR | 57.8 | 231.4 | 248.2 | 96.3 | 68.1 | 102.9 |

Table S6: Communication overhead of compared methods within one communication round.

| Method | Cost |
|---|---|
| FedProto | $p \times l \times 2$ |
| FedHPL | $l \times l \times 2$ |
| PromptFL | $m \times d \times 2$ |
| pFedPrompt | $m \times d \times 2$ |
| FedOTP | $m \times d \times 2$ |
| FedPGP | $m \times d \times 2$ |
| FedAPPR | $m \times d \times 2 + d_1 \times d_2 + d_2$ |

