# OpenReview forum: "Model-Heterogeneous Federated Prompt Learning"
_ICLR.cc/2026/Conference — ICLR 2026 Conference Withdrawn Submission_

### Official Review · Reviewer_hkVd · 2025-10-28

**Soundness:** 2
**Presentation:** 3
**Contribution:** 3
**Rating:** 4
**Confidence:** 3

**Summary:**

This paper proposes Model-Heterogeneous Federated Prompt Learning (MHFPL), a novel setting that allows clients to employ Vision-Language Models (VLMs) with diverse architectures. To address the challenge of inconsistent prompts caused by model heterogeneity, the paper introduces the FedAPPR framework. FedAPPR consists of two key components: server-level adversarial prompt alignment and client-level proximity regularization. The adversarial alignment mechanism trains a discriminator to identify whether prompts originate from a unified global distribution. The proximity regularization term constrains local client prompts to remain close to a global reference prompt. Extensive experiments demonstrate that FedAPPR outperforms existing methods under both IID and non-IID data distributions.

**Strengths:**

1. This paper is the first to investigate the problem of model heterogeneity under federated prompt learning. It aligns more closely with real-world scenarios with varying client computational resources.

2. The paper addresses the challenge of inconsistent prompt across heterogeneous models. The proposed adversarial alignment mechanism, guided by a lightweight server-side discriminator, promotes semantic consistency of prompts. Furthermore, the proximity regularization term contributes to stabilizing the overall training process.

3. The experimental evaluation is comprehensive, involving comparisons across six datasets under both IID and non-IID data partitions, which validates the superior performance of the FedAPPR framework.

**Weaknesses:**

1. Although a warm-up strategy is introduced to prevent instability from early adversarial training, the convergence curves in Figure S1 show that FedAPPR's performance improves rapidly during the warm-up phase (using only task loss and proximity regularization). The introduction of the adversarial loss does not induce a clearly discernible change in the convergence pattern. This observation raises questions about the actual contribution of the adversarial mechanism to the overall performance gain.

2. The paper operates on the assumption that historically aggregated global prompts serve as a "reliable reference." However, this assumption lacks empirical validation. Especially during early training rounds or under severe model heterogeneity, the global prompt itself may be unstable. Using it directly to train the discriminator risks leading the discriminator to learn a biased decision boundary.

3. On the client side, one optimization objective is to pull the local prompt closer to the global prompt. Yet, upon being uploaded to the server, these same "pulled-close" local prompts are subsequently labeled as "misaligned" by the discriminator for training. This creates a conflict in the optimization targets and could undermine the effectiveness of the adversarial training.

**Questions:**

1. The current experimental setup employs an identical text encoder across all clients. Would FedAPPR remain effective if the text encoders were also heterogeneous? Does the generality of the proposed method depend on a shared textual semantic space?

2. The paper lacks a fine-grained quantitative analysis of prompt similarity. Could quantitative analysis of the prompts between all clients be provided? Specifically, do clients assigned the same model architecture learn more similar prompts compared to those with different architectures?

3. In each communication round, how many training epochs are used to update the server's discriminator? Is there a specific convergence criterion for the discriminator's training, or is it trained for a fixed number of iterations?

4. FedAPPR is specifically designed for model-heterogeneous settings. How does its performance in a standard model-homogeneous federated learning scenario?

---

### Official Review · Reviewer_hGzr · 2025-10-29

**Soundness:** 3
**Presentation:** 3
**Contribution:** 2
**Rating:** 4
**Confidence:** 4

**Summary:**

Existing Federated Prompt Learning (FPL) methods rely on the unrealistic assumption of homogeneous model architectures across clients; therefore, this paper proposes the novel and practical setting of Model-Heterogeneous Federated Prompt Learning (MHFPL), where clients utilize diverse Vision-Language Models (VLMs). To tackle this MHFPL problem, the authors introduce FedAPPR (Federated Adversarial Prompt Alignment and Proximity Regularization), which integrates two key mechanisms: (1) server-level adversarial prompt alignment, which uses a discriminator to enforce cross-client semantic consistency, and (2) client-level proximity regularization, which stabilizes updates by constraining local prompts toward the global reference. Extensive experiments conducted across six benchmark datasets and diverse data distributions demonstrate that FedAPPR achieves superior performance and robustness compared to competitive baselines, affirming its effectiveness in handling heterogeneous VLMs.

**Strengths:**

1. Clear Writing and Figures: The paper clearly articulates the methodology (adversarial alignment and proximity regularization) and uses visualizations (e.g., t-SNE in Figure 1) to effectively illustrate the prompt divergence problem and the subsequent semantic coherence achieved by FedAPPR.
2. Pioneering Approach to Model Heterogeneity in FPL: This work formalizes and tackles the novel and practical challenge of Model-Heterogeneous Federated Prompt Learning (MHFPL), addressing the critical scenario where clients use diverse VLM backbones (e.g., ResNet or ViT).
3. Comprehensive Evaluation and Robustness Analysis: The evaluation is thorough, demonstrating FedAPPR's superiority across six datasets in both i.i.d. and non-i.i.d. data distributions. This analysis is strengthened by detailed ablation studies and hyperparameter sensitivity analysis on key trade-off parameters and the warm-up strategy.

**Weaknesses:**

1. Limited Technical Novelty: The core mechanisms adversarial prompt alignment​ and proximity regularization are conceptually similar to established techniques found in general Federated Learning (FL) or Model-Heterogeneous FL (MHFL) literature (e.g., using GANs for distribution matching or L2 constraints for variance reduction, similar to FedProto or FedHPL). The primary optimization target is merely switched from model weights to prompt vectors.
2. Mechanism Similarity: While ablation studies demonstrate that removing either one of them hurts performance, the fundamental conceptual similarity between regulating prompts to remain near the global reference (proximity) and pushing them toward the global distribution (adversarial alignment) raises questions about the necessity of deploying two functionally similar regularization methods concurrently.
3. Ambiguous Visual Evidence and Redundant Motivation: The primary visual evidence used to illustrate the problem and solution, Figure 1 (c) and (d), relies on t-SNE visualization. Since t-SNE is a non-linear dimension reduction technique, the claim that FedAPPR produces "more coherent and structured" prompts is heavily influenced by the projection settings and cannot serve as definitive proof of improved semantic quality independent of the final task performance. Furthermore, Figure 2, which illustrates the clear performance disparities among groups of clients using different VLM backbones (model heterogeneity), presents a generally known fact regarding varying model capacities, making its inclusion redundant as a primary motivating factor for the problem setting.
4. Constraints on Generalization and Scope:
- The paper formally introduces MHFPL to address diversity in VLM backbones (ResNet, ViT). However, the study confines its comparison to prompt-specific or VLM-specific FL baselines (e.g., PromptFL, FedHPL). To validate the model-agnostic nature of FedAPPR, the work should compare against broader model-heterogeneous FL techniques that are designed for general architectures beyond the VLM/CLIP domain, if the prompt-tuning mechanism is generally applicable outside of VLM.
- The implemented solution requires fixed dimensions: the prompt length (m=16) and prompt dimension (d=512) are set uniformly across all clients. The framework does not detail how the prompt aggregation (Eq. 4) or the server-side discriminator would accommodate a practical heterogeneous setting where client VLMs naturally utilize different prompt dimensions (d), which restricts the true extent of hardware diversity handled by the current implementation.
5. Overemphasis on Simple Mechanism: The warm-up strategy is conceptually simple. It merely delays the activation of the adversarial alignment mechanism until t>T0. While critical for stabilizing the adversarial training, devoting a specific methodology section to this simple sequential activation may unnecessarily inflate the description of the framework's core novelty.

**Questions:**

The concerns and questions are described above in the weakness section above.

---

### Official Review · Reviewer_ezPx · 2025-10-31

**Soundness:** 2
**Presentation:** 3
**Contribution:** 3
**Rating:** 4
**Confidence:** 3

**Summary:**

This paper presents Model-Heterogeneous Federated Prompt Learning (MHFPL), where clients use diverse vision-language models to collaboratively learn prompts. To tackle semantic divergence due to model heterogeneity, the authors propose FedAPPR, combining server-level adversarial prompt alignment and client-level proximity regularization. Experiments on six datasets with various model and data heterogeneity show consistent improvements over baselines. Ablation and sensitivity analyses also explore the stability and contributions of FedAPPR’s components.

**Strengths:**

1. The two-pronged approach of adversarial prompt alignment and proximity regularization is well-motivated, with intuitive and mathematical justification for resolving semantic divergence in prompt spaces.

2. The experimental evaluation is adequate, and the results convincingly support the main claims.

**Weaknesses:**

1. The rationality of the global prompt pool lacks sufficient justification. This ambiguity undermines the validity of global prompt pool as a stable, non-outdated semantic reference: retaining only the most recent rounds of historical prompts risks unstable reference distributions, while retaining all historical prompts may introduce obsolete semantics, both compromise the reliability of the adversarial alignment mechanism.

2.  Local prompts from different clients inevitably exhibit semantic disparities due to variations in their local training data. The current approach to computing the global prompt via direct data-volume-weighted averaging fails to account for these semantic differences, potentially leading to loss and deviation of semantic information.

3. The Related Work section omits a number of recent, directly pertinent papers on federated prompt tuning under model and data heterogeneity, e.g.,  FedPHA [1], pFedPT [2], etc. Could the authors elaborate on the novelty of their work by explicitly comparing it with the contributions of [1] and [2]?

[1] FedPHA: Federated Prompt Learning for Heterogeneous Client Adaptation
[2] Visual Prompt Based Personalized Federated Learning

**Questions:**

Please refer to Weaknesses for details.

---

### Note · Authors · 2025-11-25

I have read and agree with the venue's withdrawal policy on behalf of myself and my co-authors.